# Who Presents Where? A Population-Based Analysis of Socio-Demographic Inequalities in Head and Neck Cancer Patients’ Referral Routes

**DOI:** 10.3390/ijerph192416723

**Published:** 2022-12-13

**Authors:** Jennifer Deane, Ruth Norris, James O’Hara, Joanne Patterson, Linda Sharp

**Affiliations:** 1Newcastle University Centre for Cancer, Population Health Sciences Institute, Newcastle University, Newcastle-upon-Tyne NE1 4LP, UK; 2Freeman Hospital, Newcastle upon Tyne Hospitals, Newcastle-upon-Tyne NE7 7DN, UK; 3School of Health Sciences, Institute of Population Health, University of Liverpool, Liverpool L69 7ZX, UK

**Keywords:** head and neck cancer, routes to diagnosis, socio-demographic inequalities, healthcare inequalities, emergency presentation

## Abstract

Head and neck cancers (HNC) are often late stage at diagnosis; stage is a major determinant of prognosis. The urgent cancer referral pathway (two week wait; 2WW) within England’s National Health Service aims to reduce time to diagnosis. We investigated factors associated with HNC route to diagnosis. Data were obtained from the English population-based cancer registry on 66,411 primary invasive HNCs (ICD C01-14 and C31-32) diagnosed 2006–2014. Multivariable logistic regression determined the likelihood of different diagnosis routes by patients’ demographic and clinical characteristics. Significant socio-demographic inequalities were observed. Emergency presentations declined over time and 2WW increased. Significant socio-demographic inequalities were observed. Non-white patients, aged over 65, residing in urban areas with advanced disease, were more likely to have emergency presentations. White males aged 55 and older with an oropharynx cancer were more likely to be diagnosed via 2WW. Higher levels of deprivation were associated with both emergency and 2WW routes. Dental referral was more likely in women, with oral cancers and lower stage disease. Despite the decline over time in emergency presentation and the increased use of 2WW, socio-demographic variation is evident in routes to diagnosis. Further work exploring the reasons for these inequalities, and the consequences for patients’ care and outcomes, is urgently required.

## 1. Introduction

The UK lags significantly behind other European and high human development countries with regards to cancer outcomes [1]. Evidence suggests that this is due, in part, to later-stage diagnosis [2], including relatively high proportions of cancers which are diagnosed on emergency presentation [3].

In general, cancer survival rates are strongly associated with stage at diagnosis; the earlier the stage the better the chance of survival [4]. Late-stage cancer at diagnosis may be the result of delays at various points across the diagnostic pathway; these delays can be in presentation (time from symptom onset to first presentation to primary care), primary care (time from first presentation to referral for specialist assessment), and secondary care (time from specialist referral to diagnosis) [5].

### 1.1. Routes to Diagnosis in Cancer

The Urgent Cancer Pathway, known as 2 Week Wait (2WW), was established in the English National Health Service (NHS) in 2000 [6]. A target of 14 days from the point of referral for suspected cancer symptoms, to the point of first assessment with a specialist at the hospital, was put in place. Whilst, in part, this pathway was intended to reduce patient anxiety around waiting for investigations into a possible cancer diagnosis, it was also hoped that it would shorten the primary care interval, allowing identification of cancers at an earlier stage, widening treatment options, and improving survival. A decline over time in the proportion of cancers which present as emergencies has been attributed, in part, to the introduction of this pathway [7,8].

### 1.2. Head and Neck Cancers

Head and neck cancer (HNC) is an umbrella term for malignant tumours arising in the oral cavity, larynx, pharynx, nose and salivary glands. HNC is now the 8th most common cancer and is responsible for 3% of all cancer diagnosis in the UK [9]. No effective, organised, HNC screening is in place (although there are country-specific and international events designed to increase awareness among healthcare practitioners (HCP) and the public such as head and neck cancer awareness weeks). Therefore, patients are generally diagnosed due to the presence of symptoms. Symptoms vary and include ear pain, persistent sore throat, a neck lump (enlarged lymph node), persistent mouth ulcers, and airway obstruction. Due to tumour location, patients may present symptomatically at a variety of different healthcare settings, including the GP practice, community Dental practice, a Dental hospital or, less commonly, a hospital emergency department [10].

### 1.3. Inequalities in HNC

Equity in healthcare systems is a marker for healthcare quality [11]. Care should be provided in a way that does not vary in quality due to sociodemographic or socio-economic status (SES). There are multiple inequalities relating to HNC in the UK. Incidence is strongly socio-economically patterned, with rates around 2–4 times higher in those resident in more deprived, compared to less deprived areas [9]. Around 60% of HNCs are diagnosed at a late stage [12] and the proportion diagnosed early is lowest in the most deprived areas [12]. Moreover, survival is also worse in those resident in more deprived areas [13,14].

Current knowledge on the route(s) patients take to receive a HNC diagnosis is limited; improved understanding of whether there are socio-demographic inequalities in this could help to highlight areas for improvement in service provision. We therefore undertook a population-based study investigating socio-demographic inequalities in HNC routes to diagnosis in England. Specifically, this study set out to establish whether there are socio-demographic inequalities in HNC patients diagnosed via (i) emergency versus primary care routes; (ii) 2WW versus any standard primary care routes; and (iii) dentists versus all other non-emergency routes.

## 2. Methods

### 2.1. Study Design and Setting

Registrations for all patients with a primary invasive HNC (ICD C01-14 and C31-32) diagnosed in England between 2006 and 2014 were abstracted from the National Cancer Registration Database (NCRD). Ethnical approval was obtained from the Yorkshire and the Humber South Yorkshire Research Ethics Committee on 16th November 2017 (Ref number 206040), and this population-based study is reported according to the Strengthening the Reporting of Observational Studies in Epidemiology (STROBE) guidance [15].

### 2.2. Data Sources and Linkage

The NCRD is a population-based cancer registry that seeks to systematically identify and record information on all newly diagnosed tumours in patients resident in England. The registry receives data from across the NHS which approximates to around 300,000 malignant tumour diagnoses annually [16]. Reporting of NHS hospital cancer data is mandatory. Each NCRD record is linked, using patient NHS number, to UK NHS Hospital Episode Statistics (HES) data to provide information on comorbidities and cancer treatments.

A “route to diagnosis” was assigned, by the National Cancer Registration and Analysis Service (NCRAS) to each cancer registration using a combination of data from the NCRD, HES, Cancer Waiting Times, and Cancer Screening programmes. The route to diagnosis refers to a sequence of interactions between the patient and the healthcare system which leads to the diagnosis of cancer [17]. Each registration is assigned one of eight main routes to diagnosis codes: GP referral; 2WW; emergency presentation; other outpatient, screen detected (not relevant for HNC); inpatient elective; death certificate only (DCO); and unknown. Within several of these main routes, there are (sub) routes which can be used to distinguish patients who were referred from different types of practitioners (e.g., 2WW (GP), 2WW (dentist), 2WW (other)). This route to diagnosis dataset has been used to document the diagnostic route for a range of cancers [18], but it has not been previously used to compare which HNC patients are present and are diagnosed through which routes.

### 2.3. Population

The population of interest was patients with an incident primary invasive HNC (n = 70,334). In instances where a patient had records for multiple primary tumours in the head and neck (n = 1308), a hierarchy determined which tumour record to retain for analysis. This was as follows: (i) the earliest diagnosed tumour; (ii) the earliest tumour referral date; (iii) the tumour marked as potentially positive for human papilloma virus (HPV), based on proxy information (morphology and subsite); and (iv) selected at random from the remaining tumours. Childhood tumours in patients aged <20 years old were excluded from the analysis. Cases with missing routes to diagnosis (n = 2243) and cases diagnosed only at the time of death (n = 98) were also then excluded. This left an analytical cohort of 66,411 patients (Figure 1).

### 2.4. Explanatory Variables

Explanatory variables of interest were as follows: age at diagnosis, sex, cancer site, deprivation category, period of diagnosis, ethnicity, urban/rural category, stage, grade and comorbidities. Age at diagnosis was categorised as 20–54, 55–64, 65–79 and 80+ years. Cancer sites were grouped as oral cavity (C02-C06; including palate), oropharynx (C01, C09, C10), larynx (C32) and other HNC (nasopharynx C11; hypopharynx C12, C13; salivary glands C07, C08; other sites C05, C07-C08, C11-C13; and non-specific sites C14, C31). Deprivation was an area-based measure of the income domain of the Index of Multiple Deprivation (IMD) [19] Quintile 1 includes the people resident in the least deprived and quintile 5 those resident in the most deprived areas; these refer to quintiles of the general population. Deprivation was used as a SES proxy measure. Period of diagnosis was grouped into 3-year time bands (2006–2008; 2009–2011; and 2012–2014). Ethnicity was classified as white, non-white (other ethnic group) and unknown (missing and unknown ethnicity). Urban/rural categorisation was based on areas of residence at diagnosis and was collapsed to either rural or urban [20]. Cancer summary stage was assigned using the TNM staging system (I–IV or other (unknown/missing)). Tumour grade was classified as 1 (low grade, undifferentiated)—4 (high grade, differentiated) and unknown (unknown/missing). A weighted comorbidity score based on the Charlson Comorbidity Index [21] reported the number of in-patient hospital admission for different relevant comorbidities recorded in the period 3 to 27 months before diagnosis (with the index cancer disregarded). Comorbidities were classified as none, 1 and 2+.

### 2.5. Outcome Variables

The outcome variables of interest were route to diagnosis. For the purpose of this analysis, the NCRD operationalisation of (sub)route to diagnosis was categorised as follows: (i) emergency presentation (comprising (sub)routes: A&E, emergency GP referral, emergency transfer, emergency admission or attendance); (ii) all primary care routes (that is, all routes which would have been initiated in primary care: GP referral, inpatient referral, outpatient (dentist and other referral), 2WW (dentist, GP and other)); (iii) 2WW (all urgent cancer referral routes: dentist, GP and other); (iv) standard care routes (that is, all non-urgent non-emergency, cancer referral routes: GP referral, inpatient referral, outpatient (other and dentist referral)); (v) dentist (all routes which started with a dentist: outpatient and 2WW); (vi) and all other non-emergency routes (referral routes which did not start with a dentist: GP referral, inpatient, outpatient (other referral) and 2WW (GP and other)) (Appendix A).

### 2.6. Statistical Analyses

Three analyses were undertaken to explore the role of socio-demographics on route to diagnosis (Figure 1; Appendix A). Analysis 1 included the whole analytical cohort (emergency presentation) and considered whether there was a difference in those patients presenting through the emergency route compared with all primary care routes (i.e., comparing categorisations (i) and (ii) above). Analysis 2 considered only patients coming through primary care routes (category (ii) above), and compared 2WW referral versus standard primary care-initiated routes (i.e., (iii) vs. (iv)). Analysis 3 again considered only patients coming through primary routes (category (ii) above), but this time compared dentist referral vs. all other non-emergency (non-dental) routes (i.e., categories (v) vs. (vi) above).

For each analysis, baseline descriptive statistics were reported for the analytic population along with chi-square tests of associations between socio-demographic and clinical variables with diagnosis route. Univariable and multivariable logistic regression models were then developed to assess the likelihood of diagnosis route by socio-demographic characteristics with and without adjustment for confounders. Any variables significant in univariate analyses (likelihood ratio tests (LRT) *p* ≤ 0.05) were included in multivariable models. Models were reduced to contain only statistically significant variables (LRT *p* ≤ 0.05). Model goodness-of-fit was assessed, and care taken to avoid multicollinerity. The Akaike Information Criterion (AIC) was used to differentiate between competing models. All final models had adequate fit. Odds ratios (ORs) and 95% confidence intervals (CIs) were reported. Stata V.15 [22] was used for all analyses.

## 3. Results

### 3.1. Patient Characteristics

In total, 66,411 patients were diagnosed with a first, invasive primary HNC between 2006 and 2014. HNC diagnoses increased over time. Almost two-thirds of patients were aged 55–79 at diagnosis (64.1%); almost 70% were male (69.6%); and more than 4 out of 5 were of white ethnicity (81.2%). Diagnosis was associated with deprivation: a higher proportion of patients were resident in the most deprived areas (24.5%) than in the least deprived areas (15.7%). Most patients resided in an urban area at the time of diagnosis (82.4%). The site distribution was as follows: oral cavity (34.1%), larynx (23.9%), oropharynx (22.8%) and other HNC (19.2%). At diagnosis, the most common summary stage was stage IV (19.6%) and grade 2 (37.8%); stage was not recorded for most cases. Referral via each route to diagnosis was as follows: emergency (8.5%); all primary care routes (91.5%); 2WW (39.4%); standard care routes (52.1%); dentist (9.9%); and all other routes (81.6%). Demographic and clinical characteristics of the population are shown in Table 1. A full breakdown by individual route to diagnosis can be viewed in Appendix A.

### 3.2. Analysis 1: Emergency Presentation vs. All Primary Care Routes

In total, 8.5% of patients (n = 5676) were diagnosed through emergency presentation, compared to 91.5% identified through primary care (n = 60,735). The percentage of emergency presentations declined slightly over time from 9.6% in 2006–2008 to 7.9% in 2012–2014. In univariate analyses, several variables were associated with diagnosis through the emergency route. These were older age, being male, living in a more deprived area, having two or more comorbidities, non-white ethnic group, stage IV disease and higher-grade cancer (Table 2). Compared with oral cancers, cancers of the larynx and other HNCs were more likely to present through emergency routes.

Socio-demographic associations (apart from with sex) persisted in multivariable analyses and were statistically significant. Those aged 80 and over were almost twice as likely to be diagnosed through emergency presentation (80+ years old vs. 20–54 years old; multivariable odds ratio (mvOR) 2.00, 95% CI 1.82, 2.19). There was also a consistent trend of increased likelihood of emergency diagnosis as the level of deprivation increased. Those patients resident in the most deprived areas were 1.82 times more likely to come through an emergency route than those patients resident in the least deprived areas (IMD5 vs. IMD 1; mvOR 1.82, 95% CI 1.65, 2.00). Non-white patients were 1.28 times more likely to be diagnosed via emergency presentation than white patients (non-whites vs. white; mvOR 1.28, 95% CI 1.13, 1.45). Patients residing in rural areas were significantly less likely to be referred through an emergency route (rural vs. urban mvOR; 0.91, 95% CI 0.84, 0.99). In terms of clinical variables, patients diagnosed with a higher-grade cancer were 1.45 times more likely to present through emergency routes (high vs. low grade; mvOR 1.45, 95% CI 1.09, 1.93). Stage I cancers were 82% less likely than stage IV cancers to be diagnosed via emergency presentation (I vs. IV; mvOR 0.18, 95% CI 0.15, 0.23).

### 3.3. Analysis 2: 2WW vs. Standard Primary Care-Initiated Routes

Of HNC patients who were diagnosed through a route initiated in primary care, just over 40% came through the urgent 2WW pathway (n = 26,148; 43.1%). This proportion rose over time from 36.0% in 2006–2008 to 49.8% in 2012–2014. When comparing patients referred via 2WW rather than via other standard care routes, the variables associated with an increased likelihood of urgent referral in univariate analyses were as follows: being aged 55–64 years old male, and of white ethnicity; having a cancer of the oropharynx, stage III and IV disease, grade 3 tumours, no comorbidities and residing in an area of higher deprivation. There was no observed variation by urban/rural residence. In multivariable analysis, associations with stage and grade did not persist. Patients aged 55–64 years were more likely to be referred via the urgent 2WW pathway than younger patients (55–64 years vs. 20–54 years; mvOR 1.18, 95% CI 1.13, 01.24); more modest increased risks were seen for the two older age-groups. Compared to cancers of the oral cavity, cancers of the oropharynx were more likely to been referred via 2WW (mvOR 1.64, 95% CI 1.57, 1.71). Patients were 1.43 more than 40% more likely to be referred by 2WW pathways if they resided in the most deprived areas (IMD5 vs. IMD1; mvOR 1.43, 95% CI 1.35, 1.50). Being female was associated with a reduced likelihood of 2WW referral (mvOR 0.76, 95% CI 0.73, 0.79) as was being from a non-white ethnic group (non-white ethnic group vs. white; mvOR 0.57, 95%CI 0.52, 0.62) (Table 3).

### 3.4. Analysis 3: Dentist vs. All Other Non-Emergency Routes

Overall, 10.8% (n = 6572) of all HNC patients who followed a non-emergency route were referred via a dentist. This percentage rose slightly from 9.6% in 2006–2008 to 11.8% in 2012–2014. In the univariable analysis, when compared with referral via all other routes, dental referral was associated with older age, female gender, residence in a less deprived area and having an oral cancer. All variables apart from urban/rural category and comorbidities were statistically significant in the final model. In multivariable analyses, patients aged 65–79 years old were most likely to be referred via the dentist (65–79 vs. 20–54 years; mvOR 1.13, 95%CI 1.05, 1.22) as were female patients (mvOR 1.27, 95% CI 1.20, 1.34) and those from a non-white ethnic group (non-white vs. white; mvOR 1.26, 95%CI 1.12, 1.43). Residence in an area of increasing deprivation was associated with a reduced chance of dental referral when compared to all other routes (IMD 5 vs. IMD1; mvOR 0.71, 95%CI 0.65, 0.78). Patients with stage I cancer (stage I vs. stage IV; mvOR 1.19, 95%CI 1.07, 1.32) were more likely to be referred via dental routes. Diagnoses via dental referral when compared to all other routes also increased over time (2012–2014 vs. 2006–2008; mvOR 1.22, 95% CI 1.12, 1.32) (Table 4).

## 4. Discussion

To our knowledge, this is the first comprehensive analysis of routes to diagnosis for HNC in England. In this population-based study, significant socio-demographic inequalities were observed and were shown to vary across diagnosis routes.

There were some indications in the results of positive changes over time, most notably the increase in those picked up through the urgent cancer referral route (2WW). How-ever, there are several areas of concern. The analysis showed that there has been an in-crease over time in the number of HNCs diagnosed, although the distribution of HNC cancer sub-sites has changed with the predominant tumour site being the oropharynx in 2012–2014, rather than larynx which was most common in 2006–2008. This echoes trends re-ported elsewhere [23] and likely reflects changes in risk factors such as a reduction in smoking prevalence and an increase in HPV-related cancers [24]. Although overall the number of patients diagnosed through the emergency route is relatively small compared to some other cancers [25,26], there was a small increase in the number (albeit not the percentage) of emergency presentations over time. This is concerning as emergency cancer presentations may be considered, in some ways, as a “failure” of the system, and indicative of significant delays or barriers to presentation.

### 4.1. Emergency Route

Those that were diagnosed through the emergency route were more likely to present with advanced disease, which is consistent with patterns of other cancers in the UK and internationally [3]. In terms of socio-demographic characteristics, emergency presentations were more often patients from urban areas and areas of greater deprivation, from non-white ethnic groups, and over the age of 65.

The association between older age and emergency presentation is supported by previous research in all cancers in England where likelihood of emergency presentation rose significantly in those over 70 years [27]. Whilst it is known that advanced age is a risk factor for HNC, the vague or non-specific nature of some HNC symptoms may mean that symptoms are not recognised as being of concern or are perceived as “normal” aging. Previous research has shown that cancer awareness is lower in this age group than among younger people [28]. It has also shown that people, and in particular older adults, can be more reticent to seek help in primary care due to a fear of wasting clinicians’ time, particularly when symptoms are vague [29,30]. This fear may then reduce the chances of a person seeking help from primary care, resulting in a delay to diagnosis and an increased likelihood of an emergency presentation.

There are a growing number of reports on healthcare experiences of ethnic minority groups in the UK and internationally; people from ethnic minorities more often experience significant barriers to accessing healthcare, and once within the system, more often report poor experiences (for example: [31,32,33,34]). Much of the recent work has focused on the experience with COVID-19; however, it seems plausible that barriers such as lack of trust, inappropriate services and discrimination impacted help-seeking prior to COVID-19 too [35]. The finding here that patients from non-white ethnic groups are more likely to be diagnosed after an emergency presentation adds further to this accumulating evidence base [36].

Older age, deprivation and being from an ethnic minority have all been associated with suboptimal health literacy [37,38]. Health literacy is the extent to which an individual has the capacity, knowledge, understanding and confidence to access, understand, evaluate, use and navigate health and social care information and services [39]. It includes the capacity to communicate, assert and enact health decisions [40]. It has been associated—in other clinical areas—with less use of preventive health services and greater use of emergency services [39]. Given the socio-demographic patterns observed here, future research exploring the role of health literacy in emergency cancer presentation (and, more generally, across the entire cancer diagnosis pathway) would be of value.

### 4.2. Urgent Cancer Referral (2WW)

Those patients diagnosed through the urgent cancer referral route, compared to other routes which commenced in primary care, were more likely to be white, male, aged 55 years and older, resident in areas of greater deprivation and to have a cancer of the oropharynx.

The 2WW pathway requires that a patient meets a list of referral criteria for urgent investigation of a suspected cancer. Compared to other HNC tumours, oropharyngeal cancer more often presents with a neck lump/swelling [41]. This may mean that it is more likely to be recognised as potentially concerning by patients and primary care clinicians than vague, less specific (and perhaps more benign-seeming) symptoms, thus triggering a 2WW referral far quicker. Previous research on multiple different cancers has shown that those with vague symptoms delay attending primary care take a median of 34 days longer to diagnosis than those with alarm symptoms [42]. Moreover, the stereotypical “traditional” HNC patient is an older deprived male (likely with tobacco and alcohol addiction problems) [43]. It is therefore possible that primary care staff may be more likely to have a higher index of suspicion of a potentially serious underlying condition around individuals who match this profile, and therefore refer them for urgent investigation.

Some research in Denmark suggests that GPs suspect cancer in more patients than they refer onto cancer specific pathways, and that those patients who reported vague symptoms are less likely to be referred [44]. This suggests the possibility that those who do not display what the GP considers to be clear symptoms of a potential HNC, despite a suspicion of cancer, may not be being referred through the 2WW pathway.

### 4.3. Dental Referral Route

Patients from ethnic minorities, women, and those from less deprived areas, were more likely to have been referred through a dentist than through other primary care routes. As might be expected, oral cavity cancers were more often diagnosed through this route, but it is noteworthy that dentists also referred patients who were diagnosed with cancers elsewhere in the head and neck.

The dental system in the UK involves payment at the point of treatment, in contrast to the rest of primary care which is free at point of treatment; moreover, not all dental costs are subsidised by the State. There are significant barriers to accessing NHS dental services, including financial difficulties, lack of availability of services (i.e., no appointments being available), or lack of services being offered in the local area [45]. Our finding that people from more deprived areas were less likely to be diagnosed through the dental route may be explained by the cost of accessing dental check-ups and treatment. While some of those on the lowest incomes are entitled to free dental care, this involves the completion of lengthy forms [46]. Research has shown that areas of deprivation have far less NHS dentists (so called “dental deserts”) [47], suggesting that those who may be entitled to free dental care may not be able to access a dentist. This is concerning given that dentists provide a potential route for early diagnosis of some HNC.

The finding that women are more likely to have been diagnosed from a dentist referral is supported by previous research which has shown women are more likely to have made an NHS dental appointment [48]. The association between being from an ethnic minority and diagnosed through a dentist is more striking. It has been reported that people from all minority ethnicity groups have greater mistrust of dentists, are less likely to have visited a dentist and, of those who have visited, are more likely to have done so because of a specific issue rather than a routine checkup [46]. Often, research which focusses on ethnicity and health outcomes is confounded by SES, which may not be controlled for in the analysis. However, our finding was apparent after adjusting for the effects of deprivation. It is consistent with results from a small study in London, which found that once SES had been considered, Asian people were far more likely, than white people, to have visited the dentist [49]. This is an area which would benefit from further investigation.

### 4.4. Limitations

This study had several known limitations associated with analyses of routine cancer registry data. Routine data sometimes have a significant amount of missing information and in this dataset, levels of missingness for summary stage and ethnicity were high. For the latter, this meant we could not explore whether there were differences between different ethnic minority groups, and further research on this topic would be of value. For the former, care is needed in making inferences from our findings. Completeness of stage details has improved over time in registry data, so subsequent studies would be of value to confirm the findings here. We took the decision not to exclude patients with missing information as the data were unlikely to be missing completely at random, and exclusion may have introduced bias. In addition, information was not available on risk factors for HNC, such as HPV status (which was not routinely tested for during the study period), and tobacco and alcohol use; these could be associated with patient diagnostic route. The registry provided a proxy variable for inferred HPV status based on tumour site and morphology, but this was not used in the final analysis as it was not more informative than cancer site alone. It is also likely that analyses are subject to residual confounding from comorbidities; the Charlson Comorbidity Index is a crude measure of the number of comorbidities that a patient has and only includes particular conditions documented during hospital admissions in a specified time period [50], so likely underestimates true levels of comorbidity. However, as comorbidities increase with age, and age was also included in the models, any residual confounding is likely somewhat mitigated.

Another important factor is that this data are from 2006 to 2014. While cancer pathways in England have not changed in the intervening years, it is possible that the frequency with which different routes are followed, or variations between socio-demographic groups, may have changed in the intervening years. In particular, the impact of the COVID-19 pandemic had a significant impact on cancer services; in England, urgent referrals decreased dramatically, and it is estimated that there will be substantial increases in cancer deaths due to delays in diagnosis and treatment [51,52]. Research investigating whether the inequalities in route to cancer diagnosis reported here have persisted since 2014, leading up to, during, and following the pandemic should be a priority. The current analysis could usefully serve as a baseline for such future work. Finally, these results may not be generalisable to all healthcare systems outside of England which may differ in terms of processing of diagnosis routes.

## 5. Conclusions

In conclusion, this population-based analysis of English cancer registry data showed significant socio-demographic inequalities in HNC routes to diagnosis. In many instances, groups who are already experiencing higher risk of HNC are further disadvantaged by inequalities inherent within their route to diagnosis. Understanding the reasons for these inequalities is the first step to being able to improve and speed the pathways to HNC diagnosis; this, in turn, would reduce inequalities and optimise patients’ clinical outcomes.

## Figures and Tables

**Figure 1 ijerph-19-16723-f001:**
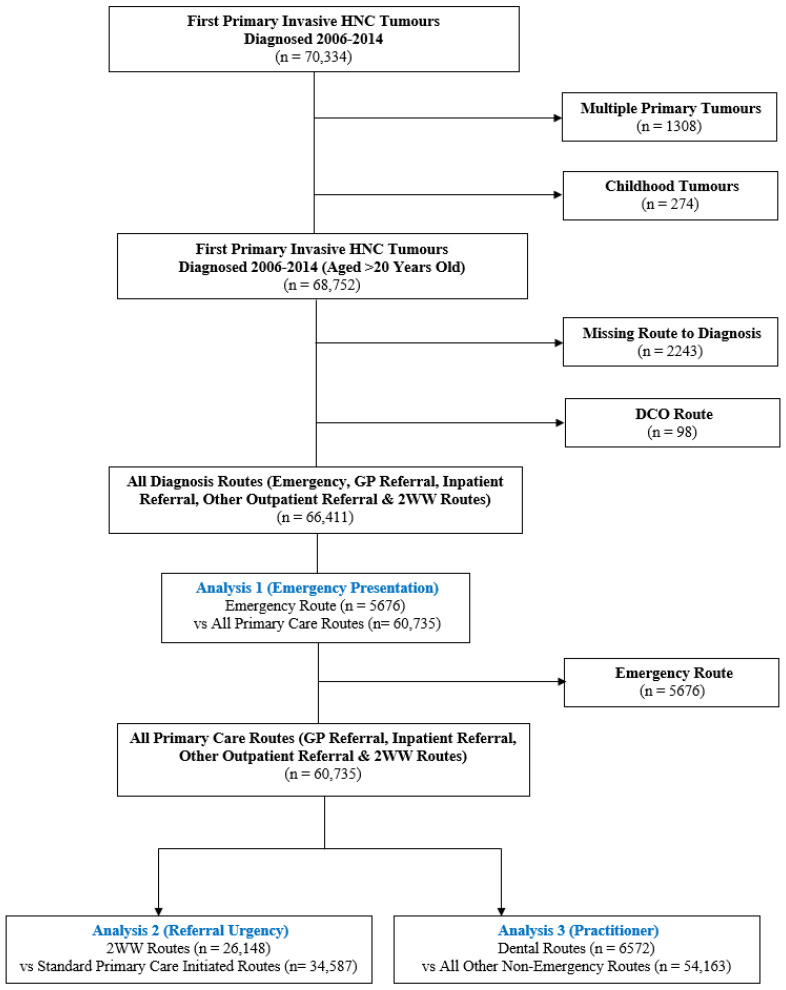
Flow diagram of the analytical cohorts for Analyses 1–3. Abbreviations: DCO: Death certificate only; GP: General practitioner; HNC: Head and neck cancer; 2WW: Two week wait.

**Table 1 ijerph-19-16723-t001:** Demographic and clinical characteristics of HNC diagnosed during 2006–2014.

	**Overall** (n = 66,411; 100%)
**Age at Diagnosis**	
20–54 years	15,259 (23.0)
55–64 years	19,459 (29.3)
65–79 years	23,092 (34.8)
80+ years	8601 (13.0)
**Sex**	
Male	46,241 (69.6)
Female	20,170 (30.4)
**Cancer Site**	
Oral Cavity ^1^	22,620 (34.1)
Oropharynx	15,128 (22.8)
Larynx	15,885 (23.9)
Other ^2^	12,778 (19.2)
**Deprivation Category**	
IMD 1 (Least Deprived)	10,417 (15.7)
IMD 2	12,260 (18.4)
IMD 3	13,245 (19.9)
IMD 4	14,217 (21.4)
IMD 5 (Most Deprived)	16,272 (24.5)
**Period of Diagnosis**	
2006–2008	19,623 (29.5)
2009–2011	22,206 (33.4)
2012–2014	24,582 (37.0)
**Ethnicity**	
White	53,919 (81.2)
Non-White ^3^	3032 (4.6)
Unknown ^4^	9460 (14.2)
**Urban/Rural Category**	
Urban	54,704 (82.4)
Rural	11,707 (17.6)
**Stage**	
I	5301 (8.0)
II	3276 (4.9)
III	3526 (5.3)
IV	13,043 (19.6)
Other ^5^	41,265 (62.1)
**Grade**	
1 (Low)	5589 (8.4)
2	25,074 (37.8)
3	17,851 (26.9)
4 (High)	603 (0.9)
Unknown ^6^	17,294 (26.0)
**Comorbidities ^7^**	
None	48,572 (73.1)
1	8690 (13.1)
2+	9149 (13.8)
**Diagnosis Route ^8^**	
2WW ^9^	26,148 (39.4)
Dentist ^10^	6572 (9.9)
Emergency ^11^	5676 (8.5)
All Other Non-Emergency Routes ^12^	54,163 (81.6)
Standard Primary Care-Initiated Routes ^13^	34,587 (52.1)
All Primary Care Routes ^14^	60,735 (91.5)

^1^ Includes palate; ^2^ Other cancer site refers to nasopharynx, hypopharynx, salivary glands, other sites and non-specific sites; ^3^ Non-white refers to other ethnic groups; ^4^ Unknown ethnicity refers to missing and unknown ethnicity; ^5^ Other stage refers to missing and unstageable tumours; ^6^ Unknown grade refers to unknown and missing tumour grades; ^7^ Measured using the Charlson Comorbidity Index; ^8^ Reported as a percentage of the analytical population (n = 66,411); ^9^ 2WW refers to 2WW (dentist), 2WW (GP) and 2WW (other); ^10^ Dentist refers to outpatient (dentist) and 2WW (dentist); ^11^ Emergency refers to A&E, emergency GP referral, emergency transfer, emergency admission or attendance; ^12^ All other non-emergency routes refers to GP referral, inpatient referral, outpatient (other referral), 2WW (GP) and 2WW (other). ^13^ Standard primary care-initiated routes refers to GP referral, inpatient referral, outpatient (other referral) and outpatient (dentist); ^14^ All primary care routes refers to GP referral, inpatient referral, outpatient (other referral), outpatient (dentist), 2WW (dentist), 2WW (GP) and 2WW (other) Abbreviations: GP: General practitioner; IMD: Index of multiple deprivation; 2WW: Two week wait.

**Table 2 ijerph-19-16723-t002:** Likelihood (OR, 95% CI and *p* values) from logistic regression of emergency versus all primary care routes by socio-demographic and clinical characteristics for Analysis 1 (n = 66,411).

	Emergency ^1^ n = 5676 (8.5%)	All Primary Care Routes ^2^ n = 60,735 (91.5%)	Analysis 1: Emergency Presentation Emergency ^1^ vs. All Primary Care Routes ^2^					
			Unadjusted			Adjusted		
			OR	95% CI	*p* Values ^3^	OR	95% CI	*p* Values ^3^
**Age at Diagnosis**					**<0.001**			**<0.001**
20–54 years	1061 (7.0)	14,198 (93.0)	1.00	-	-	1.00	-	-
55–64 years	1383 (7.1)	18,076 (92.9)	1.02	0.94–1.11	0.578	1.01	0.92–1.09	0.908
65–79 years	2006 (8.7)	21,086 (91.3)	1.27	1.18–1.38	<0.001	1.17	1.08–1.27	<0.001
80+ years	1226 (14.3)	7375 (85.7)	2.22	2.04–2.43	<0.001	2.00	1.82–2.19	<0.001
**Sex**					**0.0215**			
Male	4028 (8.7)	42,213 (91.3)	1.00	-	-	-	-	-
Female	1648 (8.2)	18,522 (91.8)	0.93	0.88–0.99	0.022	-	-	-
**Cancer Site**					**<0.001**			**<0.001**
Oral Cavity ^4^	1316 (5.8)	21,304 (94.2)	1.00	-	-	1.00	-	-
Oropharynx	1024 (6.8)	14,104 (93.2)	1.18	1.08–1.28	<0.001	1.17	1.07–1.28	0.001
Larynx	1686 (10.6)	14,199 (89.4)	1.92	1.78–2.07	<0.001	1.92	1.78–2.07	<0.001
Other ^5^	1650 (12.9)	11,128 (87.1)	2.40	2.22–2.59	<0.001	1.98	1.83–2.14	<0.001
**Deprivation Category**					**<0.001**			**<0.001**
IMD 1 (Least Deprived)	656 (6.3)	9761 (93.7)	1.00	-	-	1.00	-	-
IMD 2	808 (6.6)	11,452 (93.4)	1.05	0.94–1.17	0.371	1.06	0.96–1.19	0.254
IMD 3	1062 (8.0)	12,183 (92.0)	1.30	1.17–1.43	<0.001	1.28	1.15–1.42	<0.001
IMD 4	1319 (9.3)	12,898 (90.7)	1.52	1.38–1.68	<0.001	1.46	1.32–1.61	<0.001
IMD 5 (Most Deprived)	1831 (11.3)	14,441 (88.7)	1.89	1.72–2.07	<0.001	1.82	1.65–2.00	<0.001
**Period of Diagnosis**					**<0.001**			**0.0001**
2006–2008	1875 (9.6)	17,748 (90.4)	1.00	-	-	1.00	-	-
2009–2011	1850 (8.3)	20,356 (91.7)	0.86	0.80–0.92	<0.001	0.90	0.84–0.96	0.003
2012–2014	1951 (7.9)	22,631 (92.1)	0.82	0.76–0.87	<0.001	1.05	0.97–1.14	0.203
**Ethnicity**					**<0.001**			**<0.001**
White	4325 (8.0)	49,594 (92.0)	1.00	-	-	1.00	-	-
Non-White ^6^	318 (10.5)	2714 (89.5)	1.34	1.19–1.52	<0.001	1.28	1.13–1.45	<0.001
Unknown ^7^	1033 (10.9)	8427 (89.1)	1.41	1.31–1.51	<0.001	1.32	1.22–1.43	<0.001
**Urban/Rural Category**					**<0.001**			**0.0286**
Urban	4889 (8.9)	49,815 (91.1)	1.00			1.00		
Rural	787 (6.7)	10,920 (93.3)	0.73	0.68–0.79	<0.001	0.91	0.84–0.99	0.030
**Stage**					**<0.001**			**<0.001**
I	105 (2.0)	5196 (98.0)	0.19	0.16–0.24	<0.001	0.18	0.15–0.23	<0.001
II	115 (3.5)	3161 (96.5)	0.35	0.29–0.42	<0.001	0.30	0.25–0.37	<0.001
III	214 (6.1)	3312 (93.9)	0.62	0.53–0.72	<0.001	0.55	0.47–0.64	<0.001
IV	1235 (9.5)	11,808 (90.5)	1.00	-	-	1.00	-	-
Other ^8^	4007 (9.7)	37,258 (90.3)	1.03	0.96–1.10	0.415	0.87	0.80–0.94	0.001
**Grade**					**<0.001**			**<0.001**
1 (Low)	323 (5.8)	5266 (94.2)	1.00	-	-	1.00	-	-
2	1770 (7.1)	23,304 (92.9)	1.24	1.10–1.40	0.001	1.18	1.04–1.33	0.010
3	1424 (8.0)	16,427 (92.0)	1.41	1.25–1.60	<0.001	1.24	1.09–1.41	0.001
4 (High)	66 (10.9)	537 (89.1)	2.00	1.52–2.65	<0.001	1.45	1.09–1.93	0.012
Unknown ^9^	2093 (12.1)	15,201 (87.9)	2.24	1.99–2.53	<0.001	1.74	1.54–1.98	<0.001
**Comorbidities ^10^**					**<0.001**			**<0.001**
None	3586 (7.4)	44,986 (92.6)	1.00	-	-	1.00	-	-
1	878 (10.1)	7812 (89.9)	1.41	1.30–1.52	<0.001	1.29	1.19–1.40	<0.001
2+	1212 (13.2)	7937 (86.8)	1.92	1.79–2.05	<0.001	1.68	1.56–1.81	<0.001

^1^ Emergency refers to A&E, emergency GP referral, emergency transfer, emergency admission or attendance; ^2^ All primary care routes refers to GP referral, inpatient referral, outpatient (dentist), outpatient (other referral), 2WW (dentist), 2WW (GP) and 2WW (other); ^3^
*p* values in bold are from LRT of the contribution of the variable to the model. Unbolded *p* values are from a test of whether the OR is different from 1; ^4^ Includes palate; ^5^ Other cancer site refers to nasopharynx, hypopharynx, salivary glands, other sites and non-specific sites; ^6^ Non-white refers to other ethnic groups; ^7^ Unknown refers to missing and unknown ethnicity; ^8^ Other stage refers to missing and unstageable tumours; ^9^ Unknown grade refers to unknown and missing tumour grades; ^10^ Measured using the Charlson Comorbidity Index. Abbreviations: A&E: Accident and emergency; CI: Confidence interval; IMD: Index of multiple deprivation; GP: General practitioner; LRT: likelihood ratio test; OR: Odds ratio: 2WW; Two week wait. Model adjusted for age at diagnosis, cancer site, deprivation category, period of diagnosis, ethnicity, urban/rural categorisation, stage, grade, and comorbidities.

**Table 3 ijerph-19-16723-t003:** Likelihood (OR, 95% CI and *p* values) from logistic regression of 2WW versus standard primary care-initiated routes by socio-demographic and clinical characteristics for Analysis 2 (n = 60,735).

	2WW ^1^ n = 26,148 (43.1%)	Standard Primary Care-Initiated Routes ^2^ n = 34,587 (56.9%)	Analysis 2: Primary Care 2WW ^1^ vs. Standard Primary Care-Initiated Routes ^2^					
			Unadjusted			Adjusted		
			OR	95% CI	*p* Values ^3^	OR	95% CI	*p* Values ^3^
**Age at Diagnosis**					**<0.001**			**<0.001**
20–54 years	5983 (42.1)	8215 (57.9)	1.00	-	-	1.00	-	-
55–64 years	8415 (46.6)	9661 (53.4)	1.20	1.14–1.25	<0.001	1.18	1.13–1.24	<0.001
65–79 years	8888 (42.2)	12,198 (57.8)	1.00	0.96–1.04	0.983	1.07	1.02–1.12	0.005
80+ years	2862 (38.8)	4513 (61.2)	0.87	0.82–0.92	<0.001	1.06	1.00–1.13	0.062
**Sex**					**<0.001**			**<0.001**
Male	19,206 (45.5)	23,007 (54.5)	1.00	-	-	1.00	-	-
Female	6942 (37.5)	11,580 (62.5)	0.72	0.69–0.74	<0.001	0.76	0.73–0.79	<0.001
**Cancer Site**					**<0.001**			**<0.001**
Oral Cavity ^4^	8462 (39.7)	12,842 (60.3)	1.00	-	-	1.00		-
Oropharynx	7653 (54.3)	6451 (45.7)	1.80	1.72–1.88	<0.001	1.64	1.57–1.71	<0.001
Larynx	6152 (43.3)	8047 (56.7)	1.16	1.11–1.21	<0.001	1.07	1.02–1.12	0.005
Other ^5^	3881 (34.9)	7247 (65.1)	0.81	0.77–0.85	<0.001	0.81	0.77–0.85	<0.001
**Deprivation Category**					**<0.001**			**<0.001**
IMD 1 (Least Deprived)	3812 (39.1)	5949 (60.9)	1.00	-	-	1.00	-	-
IMD 2	4738 (41.4)	6714 (58.6)	1.10	1.04–1.16	0.001	1.11	1.05–1.17	<0.001
IMD 3	5242 (43.0)	6941 (57.0)	1.18	1.12–1.24	<0.001	1.21	1.14–1.28	<0.001
IMD 4	5649 (43.8)	7249 (56.2)	1.22	1.15–1.28	<0.001	1.26	1.19–1.33	<0.001
IMD 5 (Most Deprived)	6707 (46.4)	7734 (53.6)	1.35	1.28–1.43	<0.001	1.43	1.35–1.50	<0.001
**Period of Diagnosis**					**<0.001**			**<0.001**
2006–2008	6392 (36.0)	11,356 (64.0)	1.00	-	-	1.00	-	–
2009–2011	8494 (41.7)	11,862 (58.3)	1.27	1.22–1.33	<0.001	1.27	1.22–1.33	<0.001
2012–2014	11,262 (49.8)	11,369 (50.2)	1.76	1.69–1.83	<0.001	1.71	1.64–1.79	<0.001
**Ethnicity**					**<0.001**			**<0.001**
White	22,192 (44.7)	27,402 (55.3)	1.00	-	-	1.00	-	-
Non-White ^6^	843 (31.1)	1871 (68.9)	0.56	0.51–0.60	<0.001	0.57	0.52–0.62	<0.001
Unknown ^7^	3113 (36.9)	5314 (63.1)	0.72	0.69–0.76	<0.001	0.84	0.80–0.88	<0.001
**Urban/Rural Category**					**0.504**			-
Urban	21,478 (43.1)	28,337 (56.9)	1.00	-	-	-	-	-
Rural	4670 (42.8)	6250 (57.2)	0.99	0.95–1.03	0.504	-	-	-
**Stage**					**<0.001**			-
I	2035 (39.2)	3161 (60.8)	0.53	0.50–0.57	<0.001	-	-	-
II	1495 (47.3)	1666 (52.7)	0.74	0.69–0.80	<0.001	-	-	-
III	1733 (52.3)	1579 (47.7)	0.91	0.84–0.98	0.013	-	-	-
IV	6467 (54.8)	5341 (45.2)	1.00	-	-	-	-	-
Other ^8^	14,418 (38.7)	22,840 (61.3)	0.52	0.50–0.54	<0.001	-	-	-
**Grade**					**<0.001**			-
1 (Low)	1921 (36.5)	3345 (63.5)	1.00	-	-	-	-	-
2	10,754 (46.1)	12,550 (53.9)	1.49	1.40–1.59	<0.001	-	-	-
3	8107 (49.4)	8320 (50.6)	1.70	1.59–1.81	<0.001	-	-	-
4	209 (38.9)	328 (61.1)	1.11	0.92–1.33	0.264	-	-	-
Unknown ^9^	5157 (33.9)	10,044 (66.1)	0.89	0.84–0.95	0.001	-	-	-
**Comorbidities** ** ^10^ **					**<0.001**			**<0.001**
None	19,688 (43.8)	25,298 (56.2)	1.00	-	-	1.00	-	-
1	3342 (42.8)	4470 (57.2)	0.96	0.92–1.01	0.105	0.93	0.88–0.98	0.004
2+	3118 (39.3)	4819 (60.7)	0.83	0.79–0.87	<0.001	0.80	0.76–0.85	<0.001

^1^ 2WW refers to 2WW (dentist), 2WW (GP) and 2WW (other); ^2^ Standard primary care-initiated routes refers to GP referral, inpatient referral, outpatient (other referral) and outpatient (dentist); ^3^
*p* values in bold are from LRT of the contribution of the variable to the model. Unbolded *p* values are from a test of whether the OR is different from 1; ^4^ Includes palate; ^5^ Other cancer site refers to nasopharynx, hypopharynx, salivary glands, other sites and non-specific sites; ^6^ Non-White refers to other ethnic groups; ^7^ Unknown ethnicity refers to missing and unknown ethnicity; ^8^ Other stage refers to missing and unstageable tumours; ^9^ Unknown grade refers to unknown and missing tumour grades; ^10^ Measured using the Charlson Comorbidity Index. Abbreviations: CI: Confidence interval; IMD: Index of multiple deprivation; GP: General practitioner; LRT: Likelihood ratio tests; OR: Odds ratio: 2WW; Two week wait. Model adjusted for age at diagnosis, sex, cancer site, deprivation category, period of diagnosis, ethnicity, and comorbidities.

**Table 4 ijerph-19-16723-t004:** Likelihood (OR, 95% CI and *p* values) from logistic regression of dentist versus all other non-emergency routes by socio-demographic and clinical characteristics for Analysis 3 (n = 60,735).

	Dentist ^1^ n = 6572 (10.8%)	All Other Non-Emergency Routes ^2^ n = 54,163 (89.2%)	Analysis 3: Practitioner Dentist ^1^ vs. All Other Non-Emergency Routes ^2^					
			Unadjusted			Adjusted		
			OR	95% CI	*p* Values ^3^	OR	95% CI	*p* Values ^3^
**Age at Diagnosis**					**<0.001**			**0.0106**
20–54 years	1476 (10.4)	12,722 (89.6)	1.00	-	-	1.00	-	-
55–64 years	1784 (9.9)	16,292 (90.1)	0.94	0.88–1.02	0.119	1.04	0.97–1.13	0.283
65–79 years	2348 (11.1)	18,738 (88.9)	1.08	1.01–1.16	0.028	1.13	1.05–1.22	0.001
80+ years	964 (13.1)	6411 (86.9)	1.30	1.19–1.41	<0.001	1.06	0.96–1.16	0.256
**Sex**					**<0.001**			**<0.001**
Male	3636 (8.6)	38,577 (91.4)	1.00	-	-	1.00	-	-
Female	2936 (15.9)	15,586 (84.1)	2.00	1.90–2.11	<0.001	1.27	1.20–1.34	<0.001
Cancer Site					<0.001			<0.001
Oral Cavity ^4^	5629 (26.4)	15,675 (73.6)	1.00	-	-	1.00	-	-
Oropharynx	456 (3.2)	13,648 (96.8)	0.09	0.08–0.10	<0.001	0.11	0.10–0.12	<0.001
Larynx	43 (0.3)	14,156 (99.7)	0.01	0.01–0.01	<0.001	0.01	0.01–0.01	<0.001
Other ^5^	444 (4.0)	10,684 (96.0)	0.12	0.10–0.13	<0.001	0.13	0.11–0.14	<0.001
**Deprivation Category**					**<0.001**			**<0.001**
IMD 1 (Least Deprived)	1269 (13.0)	8492 (87.0)	1.00	-	-	1.00	-	-
IMD 2	1351 (11.8)	10,101 (88.2)	0.90	0.82–0.97	0.008	0.92	0.84–1.01	0.075
IMD 3	1354 (11.1)	10,829 (88.9)	0.84	0.77–0.91	<0.001	0.88	0.80–0.96	0.004
IMD 4	1297 (10.1)	11,601 (89.9)	0.75	0.69–0.81	<0.001	0.80	0.73–0.88	<0.001
IMD 5 (Most Deprived)	1301 (9.0)	13,140 (91.0)	0.66	0.61–0.72	<0.001	0.71	0.65–0.78	<0.001
**Period of Diagnosis**					**<0.001**			**<0.001**
2006–2008	1707 (9.6)	16,041 (90.4)	1.00	-	-	1.00	-	-
2009–2011	2202 (10.8)	18,154 (89.2)	1.14	1.07–1.22	<0.001	1.14	1.06–1.22	<0.001
2012–2014	2663 (11.8)	19,968 (88.2)	1.25	1.18–1.34	<0.001	1.22	1.12–1.32	<0.001
**Ethnicity**					**<0.001**			**<0.001**
White	5185 (10.5)	44,409 (89.5)	1.00	-	-	1.00	-	-
Non-White ^6^	407 (15.0)	2307 (85.0)	1.51	1.35–1.69	<0.001	1.26	1.12–1.43	<0.001
Unknown ^7^	980 (11.6)	7447 (88.4)	1.13	1.05–1.21	0.001	1.14	1.05–1.23	0.002
**Urban/Rural Category**					**0.0004**			
Urban	5285 (10.6)	44,530 (89.4)	1.00	-	-	-	-	-
Rural	1287 (11.8)	9633 (88.2)	1.13	1.06–1.20	<0.001	-	-	-
**Stage**					**<0.001**			**<0.001**
I	904 (17.4)	4292 (82.6)	1.74	1.59–1.91	<0.001	1.19	1.07–1.32	0.001
II	361 (11.4)	2800 (88.6)	1.07	0.94–1.21	0.319	0.88	0.77–1.01	0.073
III	273 (8.2)	3039 (91.8)	0.74	0.65–0.85	<0.001	0.83	0.72–0.96	0.013
IV	1275 (10.8)	10,533 (89.2)	1.00	-	-	1.00	-	-
Other ^8^	3759 (10.1)	33,499 (89.9)	0.93	0.87–0.99	0.027	0.89	0.82–0.96	0.005
**Grade**					**<0.001**			**<0.001**
1 (Low)	938 (17.8)	4328 (82.2)	1.00	-	-	1.00	-	-
2	3045 (13.1)	20,259 (86.9)	0.69	0.64–0.75	<0.001	0.86	0.79–0.94	0.001
3	1143 (7.0)	15,284 (93.0)	0.35	0.31–0.38	<0.001	0.62	0.56–0.68	<0.001
4 (High)	19 (3.5)	518 (96.5)	0.17	0.11–0.27	<0.001	0.54	0.33–0.87	0.011
Unknown ^9^	1427 (9.4)	13,774 (90.6)	0.48	0.44–0.52	<0.001	0.88	0.79–0.97	0.009
**Comorbidities ^10^**					**0.204**			
None	4891 (10.9)	40,095 (89.1)	1.00	-	-	-	-	-
1	802 (10.3)	7010 (89.7)	0.94	0.87–1.01	0.111	-	-	-
2+	879 (11.1)	7058 (88.9)	1.02	0.95–1.10	0.594	-	-	-

^1^ Dentist refers to outpatient (dentist) and 2WW (dentist); ^2^ All other non-emergency routes refers to GP referral, inpatient referral, outpatient (other referral), 2WW (GP) and 2WW (other); ^3^
*p* values in bold are from LRT of the contribution of the variable to the model. Unbolded *p* values are from a test of whether the OR is different from 1; ^4^ Includes palate; ^5^ Other cancer site refers to nasopharynx, hypopharynx, salivary glands, other sites, and non-specific sites; ^6^ Non-White refers to other ethnic groups; ^7^ Unknown ethnicity refers to missing and unknown ethnicity; ^8^ Other stage refers to missing and unstageable tumours; ^9^ Unknown grade refers to unknown and missing tumour grades; ^10^ Measured using the Charlson Comorbidity Index. Abbreviations: CI: Confidence interval; IMD: Index of multiple deprivation; GP: General practitioner; LRT: Likelihood ratio tests; OR: Odds ratio: Model adjusted for age at diagnosis, sex, cancer site, deprivation category, period of diagnosis, ethnicity, stage, and grade.

## Data Availability

The data used in this study were released to the authors for the purpose of this analysis; the authors are not permitted to share it. Interested individuals may apply for dataset including registrations of head and neck cancer over the study period from the current data controllers, NHS Digital.

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
