# Peer review of "Who Presents Where? A Population-Based Analysis of Socio-Demographic Inequalities in Head and Neck Cancer Patients’ Referral Routes"

_ijerph, 2022, doi:10.3390/ijerph192416723_

Round 1
Reviewer 1 Report
Dear Authors,
the manuscript is well written and brings news about head and neck cancer socio-demographic data.
I have one concern. In line 56&57 You stated that "No effective HNC screening is available..."
What about "Oral head and neck cancer awareness week" which takes place every year in April all over the world? Aaccording to my information, it is also held in English. Please add data about this event if hled in England.
Author Response
We would like to thank the reviewers for their time and feedback which was gratefully received.
In response to the comments we have made the following edits;
|
Reviewer 1 |
I have one concern. In line 56&57 you stated that "No effective HNC screening is available..."
What about "Oral head and neck cancer awareness week" which takes place every year in April all over the world? According to my information, it is also held in English. Please add data about this event if held in England.
|
Thank you for this valid point. We have added a sentence recognising that there are awareness campaigns (lines 57-59).
However, we would note that the Oral Head and Neck Cancer Awareness week is a US based event which encourages patients to go to their healthcare practitioner (HCP) for a screening session. Whilst there are similar awareness campaigns in the UK there isn’t an organised call-recall based screening programme (and that’s what we were referring to by use of the term “screening”).
|
Reviewer 2 Report
The authors present a study designed to expand current knowledge of the routes patients take to diagnosis of HNC and the extent to which sociodemographic inequalities play a role in England. Specifically, the authors wanted to determine whether socioeconomic status (SES) altered emergency vs primary care route; 2 week wait (2WW) vs any primary care routs and dental vs all other non-emergency visit routes. The authors abstracted records from the National Cancer Registry Database (NCRD) 2006-14. I agree that the knowledge expansion proposed could be useful to researchers, practitioners and policymakers alike and commend the authors for performing the study.
It is unfortunate that given this late publication date that the study has not been expanded to include more years. However, this can happen for any number for reasons: funding, work done as part of another project, etc. Do the authors intend to update the current work with more recent data? If so, when? If not, why not? Related question if the work is updated has the trend continued, is this study still relevant, have things changed since 2014 in NHS for HNC patients?
Why was HPV+/- status not included as an exploratory variable? Is it possible that this variable could have collinearity with SES/deprivation index given the link with smoking/alcohol consumption? Though you mention a lack of HPV status in the discussion you also list HPV status as a means of sorting tumor types. Please explain.
Is it reasonable to draw conclusions based on stage when over 60% of your cohort is missing this information? You mention its missingness in the discussion but not the ramifications of its missingness.
Given that most patients were over 55 years, are you certain the report of so many with no comorbidities is accurate?
The article is well written but needs the above addressed.
Author Response
We would like to thank the reviewers for their time and feedback which was gratefully received.
In response to the comments we have made the following edits;
|
Reviewer 2 |
It is unfortunate that given this late publication date that the study has not been expanded to include more years. However, this can happen for any number for reasons: funding, work done as part of another project, etc. Do the authors intend to update the current work with more recent data? If so, when? If not, why not? Related question if the work is updated has the trend continued, is this study still relevant, have things changed since 2014 in NHS for HNC patients?
|
This is a fair point and we are aware this is a limitation of this work. The original dataset was obtained for an analysis as part of JD’s PhD. Ideally, we would like to expand on this analysis further to include data from more recent years. Given that we have recently secured additional funding, an updated analysis will now be feasible. However, the new study has not yet started and the process for access to more recent data, and the costs of this, mean we cannot update these figures in a timely fashion. Despite this, we feel that the study still has merit as the pathways for referral has not changed within the NHS since 2014. Moving forward, this study provides a “baseline” from which to compare current referral patterns and routes to diagnosis. The pandemic has thrown into sharp relief cancer referral pathways in the UK; there are significant concerns about the pandemic resulting in delayed diagnoses and increasingly questions about whether the referral pathways are “fit for purpose”. We have added a sentence at the end of the limitation section to recommend that such work should be the next step (lines 456-457) and that the current analysis would provide a useful baseline. |
|
|
Why was HPV+/- status not included as an exploratory variable? Is it possible that this variable could have collinearity with SES/deprivation index given the link with smoking/alcohol consumption? Though you mention a lack of HPV status in the discussion you also list HPV status as a means of sorting tumor types. Please explain. |
We agree that HPV status could be an important confounder in this analysis, especially given associations with SES/smoking and alcohol. Unfortunately, the cancer registry does not record data on tested HPV status, and lack of such information is a limitation of this work. The cancer registry created a variable which inferred HPV status based on tumour site and morphology, and this was included in our dataset. In fitting the models, we decided that this variable of inferred HPV status did not add anything more than the cancer site variable to the multivariable models, so we did not include it. We realise, on reflection, that the way we have mentioned HPV in the manuscript is potentially misleading and that readers may conclude that we had access to HPV test information. We have revised the text in the Methods accordingly (lines 116 & 117). We have added a sentence to the limitations section to address this comment further (lines 437-441). |
|
|
Is it reasonable to draw conclusions based on stage when over 60% of your cohort is missing this information? You mention its missingness in the discussion but not the ramifications of its missingness. |
Thank you for this valid comment. We agree that the percentage of missing stage data is large, and caution is thus required with conclusions based on such staging data. Fortunately, completeness of staging has improved over time in registry data so subsequent studies would certainly be of value to confirm findings here. We have added a sentence to the limitations section regarding this (lines 431-434). |
|
|
Given that most patients were over 55 years, are you certain the report of so many with no comorbidities is accurate? |
Yes, we agree. Whilst the Charlson Comorbidity Index is a valid measure of patient comorbidities, its measurement of comorbidities is known to be somewhat crude (Charlson, 1987). As implemented in this (and other similar studies based on routinely-collected data) the measure only includes certain comorbidities occurring during in-patient hospital admissions (Schneeweiss, 2000) in a defined time window prior to the cancer diagnosis. The score therefore likely underestimates the prevalence and number of comorbidities that people actually have (e.g. they may have been admitted to hospital for other conditions, or they may have had a condition included in the index but were never admitted to hospital). This means that results could be subject to residual confounding by comorbidities. However, as the number of comorbidities a patient experiences increases with age, inclusion of age as a variable in multivariable models likely captures, to some extent, some of this residual confounding i.e. the risk estimates for age in the final models may therefore be a combination of the effect of age and residual confounding by comorbidities.
We have added additional wording to the limitations section of the manuscript to expand further on this point (lines 441-446). |
References
Charlson ME, Pompei P, Ales KL, MacKenzie CR. A new method of classifying prognostic comorbidity in longitudinal studies: development and validation. J Chronic Dis. 1987;40(5):373– 83.
Schneeweiss S, Maclure M. Use of comorbidity scores for control of confounding in studies using administrative databases. Int J Epidemiol. 2000 Oct;29(5):891–8.